# Zerumbone Ameliorates Neuropathic Pain Symptoms via Cannabinoid and PPAR Receptors Using In Vivo and In Silico Models

**DOI:** 10.3390/molecules26133849

**Published:** 2021-06-24

**Authors:** Jasmine Siew Min Chia, Ahmad Akira Omar Farouk, Tengku Azam Shah Tengku Mohamad, Mohd Roslan Sulaiman, Hanis Zakaria, Nurul Izzaty Hassan, Enoch Kumar Perimal

**Affiliations:** 1Centre for Community Health Studies (ReaCH), Faculty of Health Sciences, Universiti Kebangsaan Malaysia, Kuala Lumpur 50300, Malaysia; jasminecsm@ukm.edu.my; 2Department of Biomedical Sciences, Faculty of Medicine and Health Sciences, Universiti Putra Malaysia, Serdang 43400, Malaysia; ahmadakira@upm.edu.my (A.A.O.F.); azamshah@upm.edu.my (T.A.S.T.M.); mrs@upm.edu.my (M.R.S.); 3Department of Chemical Sciences, Faculty of Science and Technology, Universiti Kebangsaan Malaysia, Bangi 43600, Malaysia; nhz2nurhaniszakaria@gmail.com (H.Z.); drizz@ukm.edu.my (N.I.H.); 4Australian Research Council Centre of Excellence for Nanoscale BioPhotonics, University of Adelaide, Adelaide, South Australia 5005, Australia

**Keywords:** zerumbone, neuropathic pain, chronic constriction injury, cannabinoid, peroxisome proliferator-activated receptors (PPAR)

## Abstract

Neuropathic pain is a chronic pain condition persisting past the presence of any noxious stimulus or inflammation. Zerumbone, of the *Zingiber zerumbet* ginger plant, has exhibited anti-allodynic and antihyperalgesic effects in a neuropathic pain animal model, amongst other pharmacological properties. This study was conducted to further elucidate the mechanisms underlying zerumbone’s antineuropathic actions. Research on therapeutic agents involving cannabinoid (CB) and peroxisome proliferator-activated receptors (PPARs) is rising. These receptor systems have shown importance in causing a synergistic effect in suppressing nociceptive processing. Behavioural responses were assessed using the von Frey filament test (mechanical allodynia) and Hargreaves plantar test (thermal hyperalgesia), in chronic constriction injury (CCI) neuropathic pain mice. Antagonists SR141716 (CB_1_ receptor), SR144528 (CB_2_ receptor), GW6471 (PPARα receptor) and GW9662 (PPARγ receptor) were pre-administered before the zerumbone treatment. Our findings indicated the involvement of CB_1_, PPARα and PPARγ in zerumbone’s action against mechanical allodynia, whereas only CB_1_ and PPARα were involved against thermal hyperalgesia. Molecular docking studies also suggest that zerumbone has a comparable and favourable binding affinity against the respective agonist on the CB and PPAR receptors studied. This finding will contribute to advance our knowledge on zerumbone and its significance in treating neuropathic pain.

## 1. Introduction

Neuropathic pain is a chronic pain disease known worldwide as a debilitating and challenging condition to treat. Characteristic symptoms that define neuropathic pain are allodynia—pain experienced due to ordinarily non-painful stimuli, hyperalgesia—heightened pain experienced due to ordinarily painful stimuli, and dysesthesia—abnormal, painful sensation. Besides, neuropathic pain persistence lasts from months to years, affecting almost 10% of the population worldwide [1]. However, due to the complex and varying aetiology behind the development of neuropathic pain, including disease-causing nerve injuries, many individuals live through their pain undiagnosed and untreated [2]. As of now, there are no specific medications to treat neuropathic pain.

Natural products have been used for centuries for their medicinal properties. The readily consumable ginger plant, *Zingiber zerumbet*, has been documented as a folklore medicine, as it was used to treat stomachache, wounds and fever, typically in Asian countries. Zerumbone, the primary active component of *Z. zerumbet* rhizomes, has been widely studied to assess its pharmacological properties (for a review of previous studies, see [3]). Most importantly, zerumbone has shown prominent analgesic properties in acute and chronic pain animal models [4,5]. Furthermore, zerumbone has displayed no toxicity at low to moderate dosages in acute toxicity studies, indicating oral consumption safety [6,7,8].

There are multiple classes of receptors localized along the pain pathway. These receptors are either inhibitory or excitatory, depending on whether their activation amplifies nociceptive transmission or dampens it. In addition, the activation of some of these receptors works synergistically with other pain pathways. Therefore, many of these receptors are now being targeted as potential analgesics for neuropathic pain.

Cannabinoid receptor subtypes CB_1_ and CB_2_ make up the endocannabinoid system. The endocannabinoid system’s role can be dated back to almost 3000 years ago, a time when the Egyptians, Indians and Chinese are thought to have used cannabis as an analgesic [9,10,11]. However, the endogenous cannabinoid pain modulatory system was only established following the characterization of cannabinoid receptors CB_1_ [12] and CB_2_ [13], as well as the ligands anandamide [14] and 2-arachidonylglycerol (2-AG) [15,16]. 

The natural products are made up of various active compounds, consisting of natural cannabinoids and non-cannabinoid components. The rhizomes of the *Z. zerumbet* ginger plant are made up of approximately 90% sesquiterpenoids. Zerumbone, a terpene, accounts for the majority of the sesquiterpenoids found in the rhizomes of this edible ginger plant [17]. Among other terpenes found in *Z. zerumbet* are borneol, camphor, α-humulene and β-caryophyllene [18]. The sesquiterpene β-caryophyllene is also found in the *Cannabis sativa* plant. The *Cannabis sativa* plant, more commonly known as marijuana, has been reported to interact with the endogenous cannabinoid system; zerumbone is therefore postulated to interact with the system in a similar way [19].

The peroxisome proliferator-activated receptors (PPARs) are nuclear hormone receptors. They act as ligand-activated transcription factors, critical in maintaining a physiological role in regulating cellular metabolism, homeostasis and modulating gene transcription [20]. Heterodimers are formed with a retinoid X receptor (RXR) when any of the three isoforms of PPAR—α, β/δ, γ—are activated by ligands [21,22]. The significance of PPARs against inflammation was first discovered by Devchand et al. [23]. As nociception is linked to neuroinflammation, PPAR agonists’ development as analgesics for neuropathic pain has been highly encouraged.

Research on novel therapeutic agents for neuropathic pain is currently targeting cannabinoid and PPAR receptors. Both these systems have been fairly investigated in comparison to other systems involved in the pain pathway. We have elucidated the involvement of the serotonergic system in zerumbone’s anti-allodynic and antihyperalgesic effects [24], with recent findings showing the involvement of the adrenergic and opioidergic systems, TPRV, NMDA, and potassium receptors [25,26]. Further investigation should be conducted to understand zerumbone’s mechanism of action against neuropathic pain. 

Studies have shown that the inhibition of either the serotonergic or the noradrenergic system abolishes the antinociceptive effects of cannabinoid agonists [27]. Similar results also occur when the levels of NA are depleted [28]. Furthermore, cannabinoids and PPAR receptors have also been reported to act synergistically with the monoaminergic system [29,30]. For example, the cannabinoid anandamide acts most prominently on the CB_1_ and CB_2_ receptors. However, reports have shown that it operates on vanilloid receptor 1 (TRPV1) and PPARα receptors [31,32]. This evidence provides a clearer picture of various receptors and systems’ complementary effect in suppressing nociceptive transmission. A molecular docking software such as Autodock Vina may provide insights on ligand-protein interactions as well as on the molecular events occurring at the binding interface. As a strategy to further understand zerumbone’s binding interaction and affinity towards the respective receptors in this study, we employed the in silico computer-aided molecular docking method. This method is vital for complementing and supplementing the determined data [33]. Therefore, this study aims to investigate the involvement of cannabinoid and PPAR receptors in the anti-allodynic and antihyperalgesic effects of zerumbone through a neuropathic pain mice model and molecular docking studies.

## 2. Results

### 2.1. Involvement of Cannabinoid Receptors

To investigate CB_1_ and CB_2_ receptors’ involvement in the anti-allodynic effect of zerumbone, their respective antagonists SR141716 and SR144528 were pre-administered prior to zerumbone (10 mg/kg) treatment. 

Pain responses representing mechanical allodynia, as measured by the Automated Dynamic Plantar Aesthesiometer system, are displayed in Figure 1 and Figure 2. The administration of cannabinoid receptor antagonists on their own (antagonist + vehicle) is shown in Figure 1. Antagonists alone did not affect the paw withdrawal responses—not significantly different (*p* > 0.05) from those of the vehicle group. In Figure 2, regarding the S141 + Z group, zerumbone’s anti-allodynic effect decreased by 39% when CB_1_ receptors were antagonized. Comparing the S141 + Z with the zerumbone-treated group, the reduction is significantly different (*p* < 0.0001). However, in the S144 + Z group, the anti-allodynic effect of zerumbone was still present. Thus, no significant changes (*p* > 0.05) were observed when CB_2_ receptors were blocked before zerumbone treatment.

As measured by the Thermal Plantar Hargreaves Test apparatus, pain responses representing thermal hyperalgesia are displayed in Figure 3 and Figure 4. The administration of cannabinoid receptor antagonists on their own (antagonist + vehicle) is shown in Figure 3. Antagonists alone did not affect the paw withdrawal latencies—not significantly different (*p* > 0.05) from those of the vehicle group. As shown in Figure 4, the antihyperalgesic effect of zerumbone was notably (*p* < 0.0001) reversed when CB_1_ receptors were antagonized, whereas the antihyperalgesic impact was still present when CB_2_ receptors were blocked. The percentages of zerumbone’s antihyperalgesic reversal are 39% and 2% for SR141716 (CB_1_) and SR144528 (CB_2_).

### 2.2. Involvement of Cannabinoid Receptors through Molecular Docking Studies

The results of molecular docking studies using AutoDock Vina for zerumbone towards cannabinoid receptor subtypes CB_1_ and CB_2_ are summarized in Table 1. The binding position of agonist AM11542 towards CB_1_ and agonist AM841 towards CB_2_ obtained from Protein Data Bank (PDB) is used as a standard comparison for studying the binding affinity of zerumbone. The residue of the binding site interacting with zerumbone is also recorded to compare with the binding site’s residue interacting with the respective agonist used. The predicted binding mode from the docking simulation of zerumbone with the active residues of CB_1_ and CB_2_ receptors is shown in Figure 5.

### 2.3. Involvement of Peroxisome Proliferator-Activated Receptors

To investigate PPARα and PPARγ receptors’ involvement in the anti-allodynic effect of zerumbone, their respective antagonists GW6471 and GW9662 were pre-administered prior to zerumbone (10 mg/kg) treatment. 

Pain responses representing mechanical allodynia, as measured by the Automated Dynamic Plantar Aesthesiometer system, are displayed in Figure 6 and Figure 7. The administration of PPAR receptor antagonists on their own (antagonist + vehicle) is shown in Figure 6. The antagonists alone did not affect the paw withdrawal responses—not significantly different (*p* > 0.05) from those of the vehicle group. Figure 7 shows the effect of zerumbone’s anti-allodynia when PPARα and PPARγ receptors’ antagonists, GW6471 and GW966 2, respectively, were pre-administered. The anti-allodynic effect of zerumbone was significantly reversed (*p* < 0.0001) in both the G64+Z and G96+Z groups. The percentage of reversal is 51% and 46% for PPARα and PPARγ receptors, respectively.

As measured by the Thermal Plantar Hargreaves Test apparatus, pain responses representing thermal hyperalgesia are displayed in Figure 8 and Figure 9. The administration of PPAR receptor antagonists on their own (antagonist + vehicle) is shown in Figure 8. The antagonists alone did not affect the paw withdrawal latencies—not significantly different (*p* > 0.05) from those of the vehicle group. Figure 9 shows zerumbone’s antihyperalgesic effect when PPARα and PPARγ receptors’ antagonists, GW6471 and GW9662, respectively, were pre-administered. The pre-administration of GW6471 (PPARα antagonist) significantly reduced zerumbone’s antihyperalgesic impact by 40%. In contrast, the pre-administration of GW9662 (PPARγ) did not significantly (*p* > 0.05) reduce zerumbone’s antihyperalgesic effect. The percentage of reversal to zerumbone was only 4%.

### 2.4. Involvement of Peroxisome Proliferator-Activated Receptors through Molecular Docking Studies

The results of molecular docking studies using AutoDock Vina for zerumbone towards peroxisome proliferator-activated receptors (PPARs) α and γ are summarized in Table 2. The binding positions of agonist APHM13 towards PPARα and agonist JKPL53 towards PPARγ obtained from PDB are used as a standard comparison for studying the binding affinity of zerumbone. The residue of the binding site interacting with zerumbone is also recorded to compare with the binding site’s residue interacting with the respective agonist used. The predicted binding mode from the docking simulation of zerumbone with the active residues of PPARα and PPAR γ receptors is shown in Figure 10.

## 3. Discussion

Zerumbone has shown prominent pharmacological actions against both acute and chronic pain, as well as inflammation [4,5,8,34]. Among the mechanisms behind zerumbone’s action available in the literature are the involvement of the L-arginine-Nitric oxide pathways [34,35], the serotonergic system [24] and the noradrenergic [25], as well as the suppression of inflammatory mediators [36]. 

The present study demonstrated the involvement of the cannabinoid system in zerumbone’s antineuropathic effects. The endocannabinoid receptor subtype CB_1_ has been shown to play a role in zerumbone’s anti-allodynic and antihyperalgesic effects. The pre-treatment with CB_1_ receptor antagonist (SR141716, 1 mg/kg) prior to zerumbone (10 mg/kg) abolished both the anti-allodynic and antihyperalgesic effects typically observed in the chronic constriction injury animal model of neuropathic pain. The pre-administration of the CB_2_ receptor antagonist (SR144528, 1 mg/kg), on the other hand, had no significant effects on zerumbone’s anti-allodynia and antihyperalgesia.

The endocannabinoid system is made up of two distinct cannabinoid receptors: CB_1_ and CB_2_. Both these receptors are G-protein-coupled receptors. The negative coupling of cannabinoid receptors to adenylyl cyclase causes cAMP synthesis inhibition from ATP [37]. Additionally, both these receptors modulate the mitogen-activated protein kinase (MAPK) [38]. Although they are both coupled to similar G-proteins, they both activate distinct pathways to exert their physiological functions. 

The CB_1_ receptors are typically coupled with ion channels to reduce neuronal firing and inhibit neurotransmitter release. The activation of CB_1_ receptors triggers the A-type, inward potassium ion currents and suppresses N, P/Q-type calcium ion currents [39]. The activation of CB_2_ receptors does not influence potassium and calcium ions. The primary physiological function of CB_2_ receptor activation leans towards pro-survival and pro-apoptotic effects through signalling a cascade phosphatidylinositol 3-kinase and Akt (PI3K-Akt) pathway to increase the messenger ceramide synthesis [40].

Endogenous cannabinoids are retrograde neurotransmitters found post-synaptically, typically binding to cannabinoid receptors located pre-synaptically. The binding of pre-synaptic cannabinoid receptors suppresses the release of neurotransmitters. The crucial role of cannabinoid receptors is to exert an inhibitory tone in nociceptive processing. However, due to varying binding structure characteristics and the downstream signalling protein cascade, CB_1_ and CB_2_ influence different physiological systems [41]. 

Considering the most potent cannabinoid is derived from a plant, it is hypothesized that zerumbone may potentially recruit these cannabinoid receptors, either fully or partially, to exhibit its antineuropathic effects. In the present study, the endocannabinoid system’s involvement in zerumbone-induced anti-allodynia and antihyperalgesia was demonstrated through a pre-treatment with specific cannabinoid receptor antagonists. The current study found that the antagonism of CB_1_ receptors by SR141716 counteracted the antineuropathic effects of zerumbone, whereas SR144528 (CB_2_ receptor antagonist) produced no significant changes, thus implying that only CB_1_ receptors participate in zerumbone-induced anti-allodynia and antihyperalgesia.

CB_1_ receptors are expressed in both the central and peripheral nervous system. These receptors are extensively distributed in the brain, mainly in the hippocampus, cortical regions and cerebellum. Their expression in the brain appears to be responsible for the psychoactive effects of cannabinoids. Advantageously, the absence of these receptors in the medulla, thalamus and brainstem regions voids the significant life-threatening effects of potent cannabinoids [42]. The CB_1_ receptors are also expressed on presynaptic primary afferent fibres and along the axons of GABAergic interneurons [43,44]. 

The expression of CB_2_ receptors is comparably lower to that of CB_1_ receptors, as they are primarily found on immune cells [45]. Moreover, CB_2_ receptors are also abundantly expressed in dendritic cells. Therefore, the consequent expression of CB_2_ receptors on immune cells relates to their neuroinflammation and neuroprotective roles in the body’s physiological system. The discovery of the presence of CB_2_ receptors in the brain microglia and astrocyte following injury suggests that cannabinoids also act on these receptors to exert more substantial immunomodulatory effects [39].

In the present study, zerumbone was found to utilize only CB_1_ receptors to inhibit mechanical allodynia and thermal hyperalgesia. As mentioned earlier, the CB_1_ receptors are responsible for endogenous and exogenous cannabinoids’ psychoactive side effects. However, zerumbone does not cause any sedation or motor impairment [36]. Therefore, zerumbone is postulated to act as a peripherally restricted agonist to CB_1_ receptors. Peripherally restricted cannabinoids evade the psychoactive side effects associated with the activation of centrally expressed CB_1_ receptors. CB_1_ receptor agonists such as AZD1940 and AZD1704 do not cross the central nervous system and have shown promising results against inflammation and neuropathic pain-causing catalepsy [46].

In recent years, the analgesic effects of cannabinoids were reported to act synergistically with the monoaminergic system. The antinociceptive effect of WIN55,212-2, a CB_1_ receptor antagonist, was attenuated in conditions of depleted endogenous serotonin and noradrenaline levels, 5-HT_2A_-, 5-HT_7_- and α_2_-adrenergic receptor antagonisms [27,28]. CB1 receptors, which can be expressed at the amygdala, periaqueductal gray, locus coeruleus and rostral ventral medulla, are also particularly important regions of the descending monoaminergic pain pathways [47,48,49]. In this study, we found that zerumbone’s anti-allodynic and antihyperalgesic effects may not act through CB_2_ receptors. The speculated mechanism of zerumbone’s action through only one cannabinoid receptor subtype is not uncommon. The non-selective cannabinoid receptor agonist CP 55,490, for example, exhibits its antinociceptive effects via a CB_1_ receptor subtype, although it has similar affinity to both CB_1_ and CB_2_ [50,51]. 

The synergistic action between the serotonergic system and the cannabinoid system is further supported when the levels of the endogenous ligands anandamide and serotonin increase following nerve injury as an adaptive mechanism to reinforce their inhibitory tone through the descending pain pathway. CB_1_ receptors may mediate the adaptive mechanism behind the plasticity following nerve injury by increasing the dorsal raphe nucleus’ firing activity as an attempt to restore the descending inhibitory actions of serotonergic projections [29,52].

The binding interaction of zerumbone was also further studied through molecular docking simulation. Zerumbone was docked to the active site of CB_1_ and CB_2_ receptors’ crystal structures, which were retrieved from Brookhaven PDB: CB_1_ (PDB ID:5XRA) and CB_2_ (PDB ID:5ZTY). Binding energy is the primary generated parameter informing us about the strength and affinity between ligand and receptor [53]. Zerumbone has a binding affinity of 7.8/−12.7 kcal/mol towards the CB_1_ receptor and −9.4/−10.9 kcal/mol towards the CB_2_ receptor.

Zerumbone was well positioned in respect to the receptors’ binding site studied and interacted with side chains as is typical of an agonist. To look further into zerumbone’s interaction with CB_1_ receptors, the simulation found that it interacts with the side chain of Val196, Phe170, Phe174, Leu193, His178, Pro269 and Phe189—all of which, except for His178, are involved in CB_1_ agonist interactions. Similarly, the CB_2_ receptor interacts with side chains as per its agonists. It interacts primarily with Phe94, Phe87, Val113 and Phe183, all involved in CB_2_ agonist interactions. 

Although zerumbone seems to be an essential molecule for targeting receptors and enzymes, its usage remains limited because of its bioavailability problem [54]. This docking study explores the non-polar structure of zerumbone towards its interaction with the respective receptor studied. The slightly positive binding affinity value obtained in this study may reflect its unfavourable binding resulting from its non-polar structure. 

Afterwards, we investigated whether the PPAR isoforms α and γ are involved in zerumbone’s antineuropathic effects. We found that PPARα is primarily engaged in comparison to PPARγ. This was demonstrated when the pre-treatment with a PPARα antagonist, GW6471 (1 mg/kg), abolished the significant anti-allodynic and antihyperalgesic effects of zerumbone. The involvement of PPARγ, on the other hand, is only implicated in zerumbone’s anti-allodynia, as the reversal in the presence of PPARγ antagonism was only significant in response to the mechanical allodynia test, and not the thermal hyperalgesia test.

The three isoforms of PPAR (PPARα, PPARβ/δ, PPARγ) are widely expressed throughout varying tissues with different ligand specificity to exert their physiological roles. PPARs are known for their ability to alter gene expression, particularly to inhibit the expression of inflammatory genes. Furthermore, PPARs have been found throughout the nervous system, mainly on neurons and glial cells. Due to their known pathophysiological roles in conditions such as apoptosis and immunity, cumulative evidence has now shown that PPARs play a role in inhibiting the neuroinflammatory process [55,56]. 

PPARα and PPARγ are the two isoforms mainly implicated in modulatory roles in the pain pathway. PPARα isoforms are abundantly expressed on high metabolic rate tissues (e.g., heart, muscle, liver), while PPARγ receptors are found on adipose tissues [57]. Besides, PPARγ is also expressed on immune cells (e.g., macrophages, lymphocytes), whose role is to regulate macrophage activation [54,55] negatively. 

These two receptors modulate immune and inflammatory responses. In the paradigm of gene regulation, PPARα can suppress inflammatory processes by inhibiting the expression of inflammatory mediators such as tumour necrosis factor-α (TNF-α), cyclooxygenase-2 (COX-2) and prostaglandin E_2_ (PGE_2_). Moreover, the activation of PPARα inhibits the downstream signalling cascade mediated by transcription-dependent activator protein-1 (AP-1) and nuclear factor-κB (NF-κB) [58,59]. 

The first antinociceptive study on PPARα was reported in 2002, where agonists to PPARα exhibited antihyperalgesic potentials in inflammatory pain [60]. Further extensions on the antinociceptive effects of PPARα were then conducted in various animal models of nociception [61,62]. In neuropathic conditions, the PPARα agonists GW7647 and palmitoylethanolamide (PEA) managed to attenuate hyperalgesia, whereas these actions were absent in PPARα knock-out neuropathic pain mice [63].

Based on the present findings, zerumbone utilizes PPARα to exert both its anti-allodynic and antihyperalgesic effects. Apart from the gene regulatory role of PPARα, other mechanisms have been proposed regarding how these receptors can modulate nociception. Firstly, the rapid analgesic effects observed by activating PPARα may be due to its inhibitory effects on peripheral wide dynamic range neurons (WDR). The fast-acting action of PPARα on suppressing the firing of action potentials occurs by engaging significant- and intermediate Ca^2+^-activated potassium channels [63,64]. Secondly, the synergistic interaction between PPARα and the endocannabinoid system has been reported. A combinatorial, therapeutical treatment approach was observed using agonists to cannabinoid receptors and PPARα, anandamide and PEA, respectively [31,65].

Cannabinoid ligands are known to exert psychoactive effects; therefore, to avert these conditions, low dosages are preferred. However, low dosages of cannabinoid ligands, synthetic or endogenous, are typically ineffective. Interestingly, Russo et al. [31] demonstrated that low doses of the cannabinoid agonist anandamide, co-administered with a PPARα agonist, synergistically exhibited antinociception in an animal model of acute pain. Concerning this finding, zerumbone was previously shown to utilize the endogenous cannabinoid system, primarily the CB_1_ receptors, to exhibit its antineuropathic effects in the chronic constriction injury-induced neuropathic pain mice. To conform to the known synergistic interaction between the two systems, PPARα also participates in exhibiting zerumbone’s anti-allodynic and antihyperalgesic effects.

Regarding PPARγ, the anti-allodynic and antihyperalgesic actions of peroxisome ligands differ based on these receptors’ site. The mechanism of action of PPARγ in the brain produces rapid negative feedback on nociceptive transmission [66]. Recent studies have discovered the expression of PPARγ along the pain pathway, mainly on dorsal root ganglions and the spinal cord dorsal horn. Therefore, in some studies, PPARγ ligands only exhibited either or both anti-allodynic and antihyperalgesic effects. 

Taking the study by Morgenweck et al. [66] as an example, the authors found that the PPARγ agonists rosiglitazone and 15d-PGJ2 managed to attenuate nociceptive transmission only through direct stimulation of PPARγ in the brain. A systemic administration of the PPARγ ligands did not manage to elicit similar antihyperalgesic effects. Therefore, they have suggested that transcription-independent mechanisms are involved in the anti-allodynic actions of PPARγ activation. This was later confirmed by Maeda et al. [67] and Churi et al. [68], indicating that spinal sites of PPARγ are crucial for its ligands to elicit anti-allodynia. 

Maeda et al. [67] hypothesize that pioglitazone’s PPARγ activation (PPARγ agonist) elicits antihyperalgesic effects by acting on glial cells, whereas the anti-allodynic effects primarily contribute to its action on microglia and astrocytes. Moreover, the activity of PPARγ on allodynia is proposed to occur at the spinal level following nerve injury, although the underlying mechanisms are still unclear. The current findings indicate that PPARγ is involved in zerumbone’s anti-allodynic effect and not its antihyperalgesic effect. With regard to reports on PPARγ mechanisms, it is hypothesized that zerumbone utilizes only spinally expressed PPARγ

Additionally, the binding affinity of zerumbone towards PPARα and PPARγ found in the present study was −5.1/−12.2 kcal/mol and −6.1/−10.0 kcal/mol, respectively. Zerumbone was docked to the active site of PPARα and PPARγ crystal structures retrieved from Brookhaven PDB: PPARα (PDB ID:3VI8) and PPARγ (PDB ID:3VJI). Zerumbone interacts with the side chain of Ala333, Val332, Cys275, Ile339, Cys276 and Ile272, which are involved in PPARα agonist interaction. A similar observation was also seen with PPARγ agonist interaction, with side chains Arg288, Ile341, Leu333, Leu330. 

To conclude, it is noteworthy that all these specific binding interactions formed between zerumbone and the receptors studied involve those similar essential amino acids responsible for agonist interactions. From the docking results, zerumbone displays a comparable and favourable binding energy to agonists that bind to each respective receptor. The result above suggests the potential activity of zerumbone towards all receptors studied, thus supporting the hypothesis that zerumbone may potentially recruit cannabinoid and PPAR receptors, either fully or partially, to exhibit its effects.

## 4. Materials and Methods

### 4.1. Zerumbone

The zerumbone used for this study was previously extracted by Sulaiman et al. [4]. Zerumbone was freshly prepared for each experiment by dissolving with dimethylsulfoxide (DMSO), Tween20 and 0.99% NaCl (normal saline) at a ratio of 5:5:90.

### 4.2. Animals

Male ICR mice, aged 6–8 weeks, were used for this study. Mice weighing 25–35 g were selected and housed under a 12 h light/dark cycle, at 24 ± 1 °C, with access to food and water ad libitum. The animals were allowed to acclimatize to the laboratory at least one week before the experimental procedures and were only used once. All procedures adhered to the Ethical Guidelines for Investigation of Experimental Pain in Conscious Animals by the International Association for the Study of Pain (IASP) [69], with approval by the Institutional Animal Care and Use Committee (IACUC) of Universiti Putra Malaysia (UPM) (Ref: UPM/IACUC/AUP-R060/2013).

### 4.3. Neuropathic Pain Induction

The chronic constriction injury surgery described by Bennett and Xie [70] was selected as the animal model to induce neuropathic pain, with slight modifications [5,36,71,72,73]. Surgical procedures were conducted while the animals were under tribromoethanol (250 mg/kg, i.p.) anaesthesia in aseptic conditions. A small incision on the left hind limb was dissected, and the skin and muscle layers were separated. A blunt dissection was made between the gluteus superficialis and biceps femoris to expose the left sciatic nerve. Peripherally-induced neuropathic pain was caused by placing one loose ligature (4-0 braided silk suture, DemeTech Sutures, Miami, FL, USA) on the sciatic nerve [72]. A BRILON non-absorbable surgical suture (Vigilenz Medical Devices Sdn. Bhd., Selangor, Malaysia) was used to suture the skin with the external application of iodine. Sham-operated animals received similar surgical procedures but without the ligature to the sciatic nerve. The animals were left to recover and develop neuropathic pain for 14 days. 

### 4.4. Assessment of Mechanical Allodynia

The Automated Dynamic Plantar Aesthesiometer system (Ugo Basile, Varese, Italy), modified from the manual von Frey filament tests by Chaplan et al. [74], was used to measure mechanical allodynia. The animals were acclimatized in clear boxes, placed on a raised wire-mesh podium. The touch-stimulator unit with a thin steel filament was placed below the left hind paw’s mid-plantar surface. With an automated increase in force, the filament was lifted against the plantar surface. A maximum force was recorded once the animal withdrew its paw. The cut-off point was set to a leading force of 5 g within 20 s.

### 4.5. Assessment of Thermal Hyperalgesia

The Hargreaves Plantar Test apparatus (Ugo Basile, Italy), based on Hargreaves, et al. [75], was used to measure thermal hyperalgesia, with slight modifications [26,76]. The animals were allowed to acclimatize in individual clear boxes placed on a raised glass podium. The mobile infrared heat source was placed below the mid-plantar surface of the left hind paw. The time taken was recorded once the animal withdrew its paw against the heat source. The cut-off point was set at 20 s.

### 4.6. Drugs and Chemicals

Tween20, DMSO, tribromoethanol, amitriptyline and antagonists SR141716, GW6471 and GW9662 were purchased from Sigma-Aldrich Chemical Co. (St. Louis, MO, USA). Antagonist SR144528 was purchased from Cayman Chemical Co., Ann Arbor, MI, USA. Amitriptyline was dissolved in normal saline (0.99% NaCl). The vehicle consisted of DMSO, Tween 20 and normal saline at a ratio of 5:5:90. All treatments were administered intraperitoneally, in a volume of 10 mL/kg.

### 4.7. Experimental Design

To determine whether zerumbone utilizes cannabinoid and PPAR receptors, the antagonists were pre-administered prior to zerumbone treatment. The animals were subjected to CCI surgery and left to recover and develop neuropathic pain for 14 days. The antagonists were administered to the animals (*n* = 6) on day 14. A vehicle or zerumbone (10 mg/kg) was administered 30 min after the antagonists. Behavioural tests were conducted 30 min following the last respective treatment administration.

To investigate the involvement of cannabinoid receptors, the antagonists used were SR141716, 1 mg/kg (CB_1_) and SR144528, 1 mg/kg (CB_2_). The antagonists and their dosages were selected based on previous studies [65,77]. SR141716 and SR144528 were dissolved in DMSO and normal saline at a ratio of 5:95. 

To investigate the PPAR receptors’ involvement, the antagonists used were GW6471, 1 mg/kg (PPARα) and GW9662, 1 mg/kg (PPARγ). The antagonists and their dosages were selected based on previous studies [65,78,79]. GW6471 and GW9662 were dissolved in ethanol, Tween80 and normal saline at a ratio of 1:1:8.

### 4.8. Molecular Structure Preparation

The protein crystal structure of CB_1_ (PDB ID:5XRA), CB_2_ (PDB ID:5ZTY), PPARα (PDB ID:3VI8) and PPARγ (PDB ID:3VJI) receptors with inhibitors was retrieved from PDB (available at http://www.rcsb.org, accessed on 20 December 2018) with the resolutions 2.80 Å, 2.80 Å, 1.75 Å, 2.61 Å respectively. The protein crystal structure was pre-treated before the docking process. The respective ligands, non-protein molecules and water molecules were removed and employed as receptors for the docking analysis. Discovery Studio 2.5.5 (Accelrys, Inc., San Diego, CA, USA) was used for the above protein and ligand preparation. For the ligand structure preparation, the 2D structure of zerumbone was drawn in the Chem3D 15.0 programme, and the energy was minimized. Then, the 2D ligand structure was converted to PDB files using Discovery Studio 2.5.5.

### 4.9. Molecular Docking

The PDB files were prepared for docking in AutoDock Vina. Each of the protein crystal structures was loaded and converted to pdbqt format. Hydrogen and partial charges were added, and the target site was assigned along with the dimension. All structures were saved in pdbqt format. The ligand and zerumbone were also loaded and converted into pdbqt format. The best free energy of the binding values was later obtained. The docking visualization of zerumbone and each receptor were visualized in Discovery Studio 2.5.5 [80].

### 4.10. Statistical Analysis

Data are displayed as mean ± standard error of the mean (SEM). A One-Way ANOVA followed by a Tukey’s post hoc test using the GraphPad Prism v6.0 software (GraphPad, San Diego, CA, USA) were used to analyse behavioural data. The level of significance was set at *p* < 0.05.

## 5. Conclusions

In summary, the present study demonstrates for the very first time that zerumbone utilizes cannabinoid CB_1,_ PPARα and PPARγ to elicit its anti-allodynic and antihyperalgesic effects through both the in vivo and in silico models used. Our findings potentiate the antinociceptive property of zerumbone by further describing its mechanisms of action, with favourable binding energy towards cannabinoid and PPAR receptors. As we highlight the significance of cannabinoid and peroxisome proliferator-activated receptors, this study demonstrates the involvement of multiple mechanisms of action of zerumbone against neuropathic pain.

## Figures and Tables

**Figure 1 molecules-26-03849-f001:**
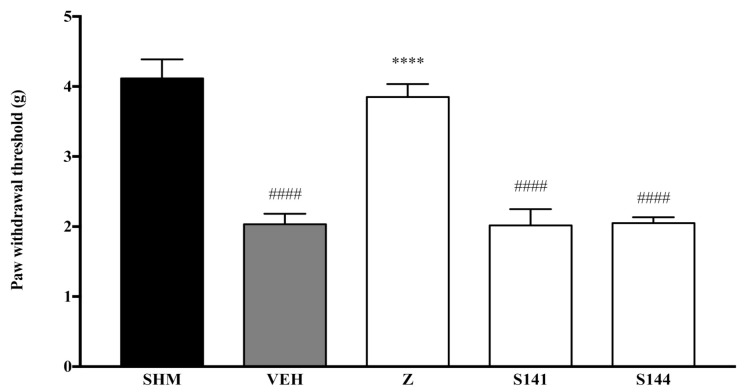
The effect of cannabinoid receptor antagonists’ co-administration with a vehicle on mechanical allodynia in CCI-induced neuropathic pain mice. Each column represents the mean ± SEM, *n* = 6 mice per group. ^####^
*p* < 0.0001 as compared to the sham group and **** *p* < 0.0001 as compared to the vehicle group. Sham (SHM), Vehicle (VEH: 10 mL/kg, i.p.), Zerumbone (Z: 10 mg/kg, i.p.), Antagonists; SR141716, CB_1_ (S141: 1 mg/kg, i.p.), SR144528, CB_2_ (S144: 1 mg/kg, i.p.).

**Figure 2 molecules-26-03849-f002:**
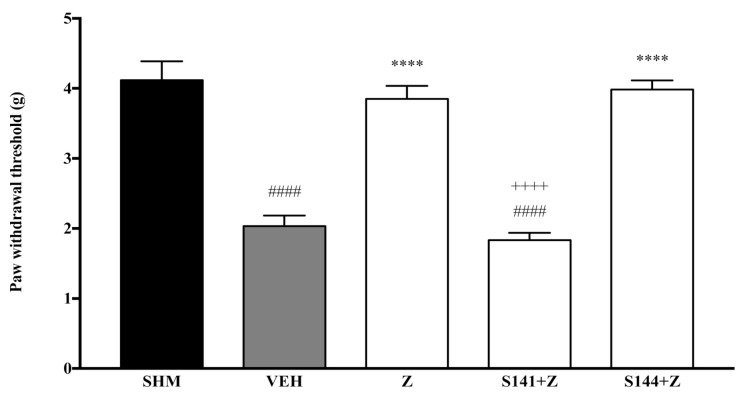
The effect of cannabinoid receptor antagonists’ co-administration with zerumbone on mechanical allodynia in CCI-induced neuropathic pain mice. Each column represents the mean ± SEM, *n* = 6 mice per group. ^####^ *p* < 0.0001 as compared to the sham group, **** *p* < 0.0001 as compared to the vehicle group and ^++++^ *p* < 0.0001 as compared to the zerumbone-treated group. Sham (SHM), Vehicle (VEH: 10 mL/kg, i.p.), Zerumbone (Z: 10 mg/kg, i.p.), Antagonists; SR141716, CB_1_ (S141: 1 mg/kg, i.p.), SR144528, CB_2_ (S144: 1 mg/kg, i.p.).

**Figure 3 molecules-26-03849-f003:**
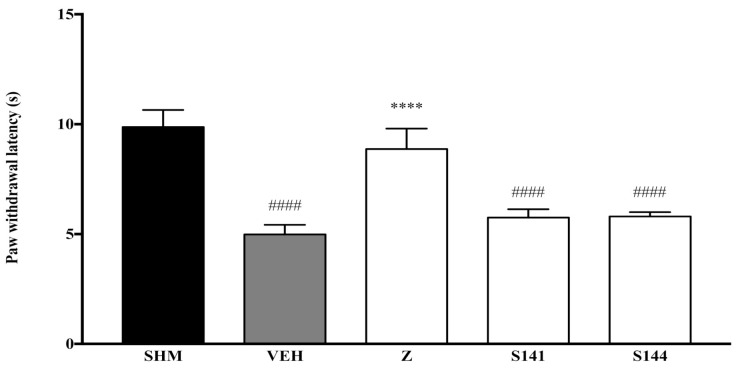
The effect of cannabinoid receptor antagonists’ co-administration with a vehicle on thermal hyperalgesia CCI-induced neuropathic pain mice. Each column represents the mean ± SEM, *n* = 6 mice per group. ^####^ *p* < 0.0001 as compared to the sham group and **** *p* < 0.0001 as compared to the vehicle group. Sham (SHM), Vehicle (VEH: 10 mL/kg, i.p.), Zerumbone (Z: 10 mg/kg, i.p.), Antagonists; SR141716, CB_1_ (S141: 1 mg/kg, i.p.), SR144528, CB_2_ (S144: 1 mg/kg, i.p.).

**Figure 4 molecules-26-03849-f004:**
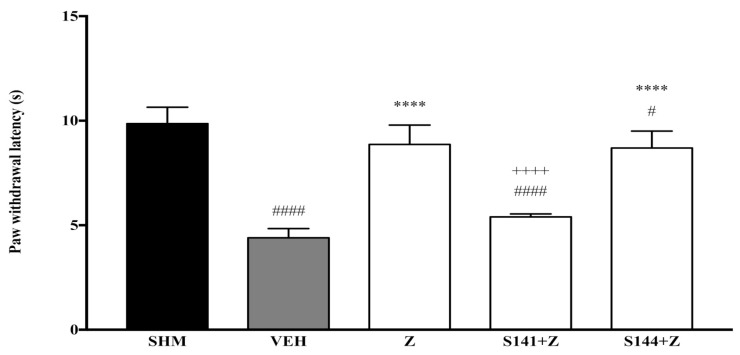
The effect of cannabinoid receptor antagonists’ co-administration with zerumbone on thermal hyperalgesia CCI-induced neuropathic pain mice. Each column represents the mean ± SEM, *n* = 6 mice per group. ^#^ *p* < 0.05, ^####^ *p* < 0.0001 as compared to the sham group, **** *p* < 0.0001 as compared to the vehicle group and ^++++^ *p* < 0.0001 as compared to the zerumbone-treated group. Sham (SHM), Vehicle (VEH: 10 mL/kg, i.p.), Zerumbone (Z: 10 mg/kg, i.p.), Antagonists; SR141716, CB_1_ (S141: 1 mg/kg, i.p.), SR144528, CB_2_ (S144: 1 mg/kg, i.p.).

**Figure 5 molecules-26-03849-f005:**
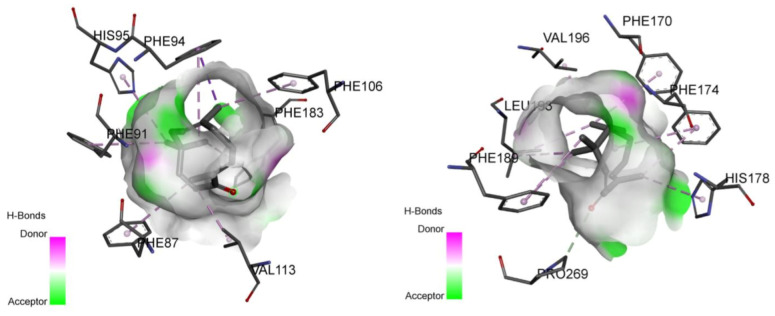
Predicted binding mode from the docking simulation of zerumbone with the active residues of CB_1_ (**left**) and CB_2_ (**right**) receptors.

**Figure 6 molecules-26-03849-f006:**
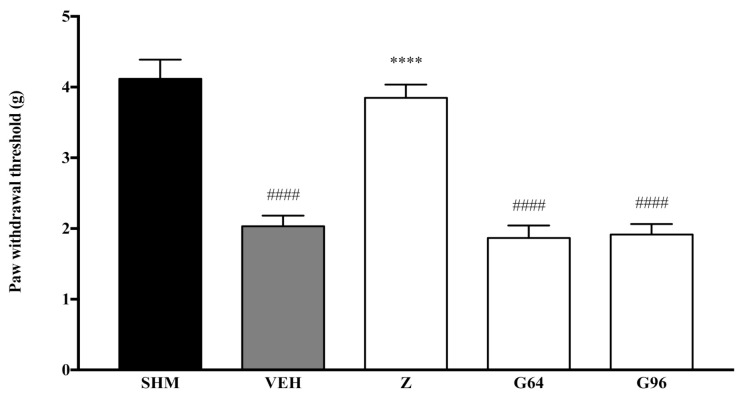
The effect of PPAR antagonists’ co-administration with a vehicle on mechanical allodynia in CCI-induced neuropathic pain mice. Each column represents the mean ± SEM, *n* = 6 mice per group. ^####^
*p* < 0.0001 as compared to the sham group and **** *p* < 0.0001 as compared to the vehicle group. Sham (SHM), Vehicle (VEH: 10 mL/kg, i.p.), Zerumbone (Z: 10 mg/kg, i.p.), Antagonists; GW6471, PPARα (G64: 1 mg/kg, i.p.), GW9662, PPARγ (G96: 1 mg/kg, i.p.).

**Figure 7 molecules-26-03849-f007:**
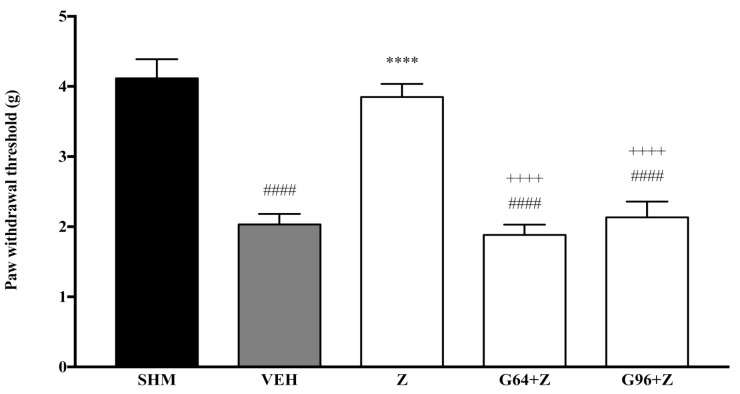
The effect of PPAR antagonists’ co-administration with zerumbone on mechanical allodynia in CCI-induced neuropathic pain mice. Each column represents the mean ± SEM, *n* = 6 mice per group. ^####^
*p* < 0.0001 as compared to the sham group, **** *p* < 0.0001 as compared to the vehicle group and ^++++^
*p* < 0.0001 as compared to the zerumbone-treated group. Sham (SHM), Vehicle (VEH: 10 mL/kg, i.p.), Zerumbone (Z: 10 mg/kg, i.p.), Antagonists; GW6471, PPARα (G64: 1 mg/kg, i.p.), GW9662, PPARγ (G96: 1 mg/kg, i.p.).

**Figure 8 molecules-26-03849-f008:**
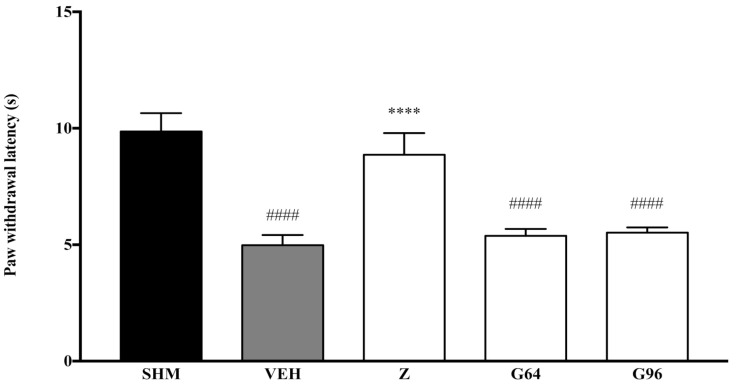
The effect of PPAR antagonists’ co-administration with vehicle thermal hyperalgesia in CCI-induced neuropathic pain mice. Each column represents the mean ± SEM, *n* = 6 mice per group, ^####^
*p* < 0.0001 as compared to the sham group and **** *p* < 0.0001 as compared to the vehicle group. Sham (SHM), Vehicle (VEH: 10 mL/kg, i.p.), Zerumbone (Z: 10 mg/kg, i.p.), Antagonists; GW6471, PPARα (G64: 1 mg/kg, i.p.), GW9662, PPARγ (G96: 1 mg/kg, i.p.).

**Figure 9 molecules-26-03849-f009:**
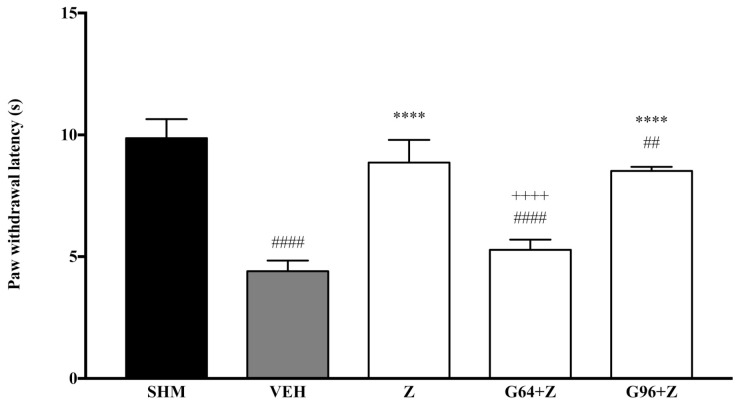
The effect of PPAR antagonists’ co-administration with zerumbone on thermal hyperalgesia in CCI-induced neuropathic pain mice. Each column represents the mean ± SEM, *n* = 6 mice per group. ^##^
*p* < 0.01, ^####^
*p* < 0.0001 as compared to the sham group, **** *p* < 0.0001 as compared to the vehicle group and ^++++^
*p* < 0.0001 as compared to the zerumbone-treated group. Sham (SHM), Vehicle (VEH: 10 mL/kg, i.p.), Zerumbone (Z: 10 mg/kg, i.p.), Antagonists; GW6471, PPARα (G64: 1 mg/kg, i.p.), GW9662, PPARγ (G96: 1 mg/kg, i.p.).

**Figure 10 molecules-26-03849-f010:**
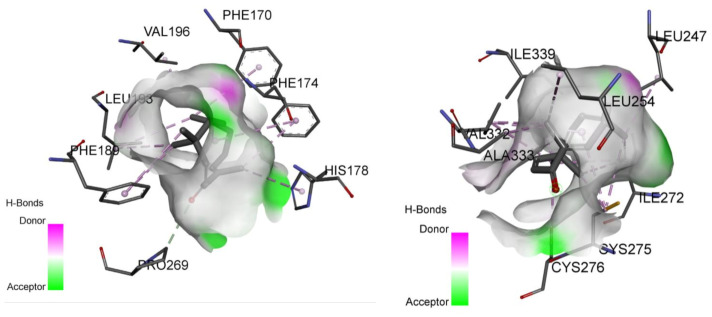
Predicted binding mode from the docking simulation of zerumbone with the active residues of PPARα (**left**) and PPARγ (**right**) receptors.

**Table 1 molecules-26-03849-t001:** Binding affinity (kcal/mol) of the favourable conformation based on AutoDock Vina and the amino acids involved during the respective interaction.

Best Binding Energy (kcal/mol) for		Ligands		
	Zerumbone	Amino Acid Residue	Agonists CB_1_ (AM11542), CB_2_ (AM841)	Amino Acid Residue
CB_1_ receptor	−7.8	Val196, Phe170, Phe174, Leu193, His178, Pro269, Phe189	−12.7	Phe177, Phe174, Phe170, Phe379, Ser383, Cys386, Leu359, Phe200, Val196, Leu193, Phe268, Pro269, Phe189, Ile271, Leu276, Tyr275
CB_2_ receptor	−9.4	His95, Phe94 Phe91, Phe87 Val113, Phe 183 Phe106	−10.9	Phe94, Ile110, Phe87, Val113, Phe183, Phe117, Cys288, Phe281, Val261, Met265, Trp258

**Table 2 molecules-26-03849-t002:** Binding affinity (kcal/mol) of the favourable conformation based on AutoDock Vina and the amino acids involved during the respective interaction.

Best Binding Energy (kcal/mol) for		Ligands		
	Zerumbone	Amino Acid Residue	Agonists PPARα (APHM13), PPARγ (JKPL53)	Amino Acid Residue
PPARα receptor	−5.1	Leu254, Ala333, Val332, Cys275, Ile339, Cys276, Ile272, Leu247	−12.2	His 440, Tyr 464, Tyr 314, Ser280, Cys276, Phe273, Ile272, Cys275, Cys276, Leu321, Val255, Val332, Ala333, Ile339
PPARγ receptors	−6.1	Leu333, Val339, Met364, Cys285 Arg288, Ile341, Leu330, Ile326	−10.0	Tyr473, Gly284, Phe282, Arg280, Arg288, Cya285, Tyr327, Ile341, Leu333, Leu330

## Data Availability

The data presented in this study are available on request from the corresponding author.

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
