# Peer review of "Zerumbone Ameliorates Neuropathic Pain Symptoms via Cannabinoid and PPAR Receptors Using In Vivo and In Silico Models"

_molecules, 2021, doi:10.3390/molecules26133849_

Round 1

Reviewer 1 Report

The manuscript describes attempts to identify molecular mechanisms in anti-allodynia and antihyperalgesia effects of zerumbone using in vivo animal tests in the presence and absence of known CBR and PPAR antagonists. The design of the study is questionable since in vitro receptor binding or functional experiments with CBR and PPAR would answer questions about interactions of zerumbone with these receptors much better and without any need to sacrifice animals. The functional in vivo results open many rooms for interpretation and do not give direct answer these questions due complexity of biochemistry behind the pain phenomenon. Moreover, the obtained results appear contradictory. If both CBR and PPAR systems are involved in anti-allodynia and antihyperalgesia effects of zerumbone, why zerumbone was completely inactive after blocking only one of them: selectively CB1 only or selectively PPARalfa or PPARgamma only? Authors should discuss and explain such observations. Without results of in vitro binding experiments, authors' conclusion sounds too speculative. Poor English language is another serious issue of this manuscript. An extensive editing of English language and style in required. I believe that the language problem results in statements like: "The Z. zerumbet ginger plant's rhizomes are made up of almost 90% sesquiterpenoids, along with trace amounts of roughly 30 other compounds. Zerumbone accounts for 70% of the sesquiterpenoids found in the rhizomes of this edible ginger plant [17]. The sesquiterpene β-caryophyllene accounts for approximately 40% of the Cannabis sativa plant." These % values probably refer to some extracts rather than whole part of plants. "In Figure 2, the S141+Z group, zerumbone's anti-allodynic effect reduced by 39% when CB1 receptors were antagonized." From Figure 2, the effect is completely (~100%) reversed to values of the vehicle group. The same is applied for all figures discussing the interaction of zerumbone with CBR and PPAR antagonists.

Author Response

We would like to sincerely thank the editor and reviewer for their thorough comments and the time they have taken to provide suggestions in improving the quality of this manuscript. We appreciate the feedback, and our responses are as attached.

Reviewer 2 Report

On the manuscript " Zerumbone suppresses allodynia and hyperalgesia via cannabinoid and PPAR receptors in chronic constriction injury mice model of neuropathic pain and molecular docking studies "  by Jasmine Siew Min Chia et al., the authors investigated the involvement of cannabinoid and PPAR receptors in the anti-allodynia and anti- hyperalgesic effects of zerumbone through neuropathic pain mice model and molecular docking studies. The study demonstrated that zerumbone utilizes cannabinoid CB1, PPARα and PPARγ to elicit its anti-allodynic and antihyperalgesic effects allowing describing its mechanisms of action, with favourable 585 binding energy towards cannabinoid and PPAR receptors.

The paper fit the aims and scope of the Molecules.

SOME COMMENTS

The work is technically sound and scientifically valid. The used methodology is appropriate and the objectives of fully supported by the data presented and the claims are appropriately discussed in the context of previous literature. In addition, the paper provided sufficient methodological detail that the experiments can be reproduced. the work in addition to the experimental design are well planned and followed. The conclusions drawn are supported by the data.

I suggest the addition of an Abbreviation list since throughout tte manuscript there are a lot of abbreviations

I would like suggest it for publication after the minor modifications.

TITLE

The title is too long, should be concise and informative. Do not include abbreviations or trade names in the title should be avoided. So, I suggest that the title be altered as following “ Towards elucidation the mechanisms underlying zerumbone's antineuropathic actions” or close to this

ABSTRACT

Abstract is adjusted to the developed work, to the used methodologies and to the obtained results.

INTRODUCTION

The introduction is interesting, well designed and structured, and adjusted to the target subject .

MATERIALS AND METHODS

The used methodologies and approaches were well designed, performed and described. RESULTS AND DISCUSSION

The obtained results are very interesting, are well presented and discussed.

In my opinion this paper achieved the scientific quality standards needed to be published in Molecules, after the required revisions.

Author Response

(The authors gave the same response as above.)

Reviewer 3 Report

In my opinion, in the presented form the manuscript (molecules-1208306) entitled ‘Zerumbone suppresses allodynia and hyperalgesia via cannabinoid and PPAR receptors in chronic constriction injury mice model of neuropathic pain and molecular docking studies’ described by Jasmine Siew Min Chia, Ahmad Akira Omar Farouk, Tengku Azam Shah Tengku Mohamad, Mohd Roslan Sulaiman, Hanis Zakaria, Nurul Izzaty Hassan and Enoch Kumar Perima can be recommended for publication in Molecules after minor revision.

My remarks and recommendations to the Molecules are as follows:

The text is comprehensible.

 Conclusions

After reading the publication, I have doubts about the scientific novelty of the scientific research of the reviewed manuscript. Therefore, the Authors should clearly indicate what is the scientific novelty of their research.

Please indicate clearly what is new with your manuscript (Conclusions) for the Molecules, especially in comparison to earlier of publication(s).

Author Response

(The authors gave the same response as above.)

Round 2

Reviewer 1 Report

Unfortunately, not all concerns raised at the initial review of the manuscript have not been adequately addressed by the authors.  

Receptor binding in vitro experiments is the missing link between in silico docking exercises and in vivo testing. 

The desription of the results does not correspond to the diagrams.

The style and grammar issuses mainly remain and a couple of new were added by text modifications (including the one in the abstract).

I cannot reccommend this namuscript for publication.